# OFFLINE REINFORCEMENT LEARNING VIA ACTION-SPACE PSEUDO-LABELING

## ABSTRACT

The critical challenge of offline reinforcement learning (offline RL) is improving from a fixed dataset while avoiding overestimation on out-of-distribution (OOD) actions. Existing methods typically regularize the learned policy to avoid choosing overestimated OOD actions. However, we argue that this often over-constrains policy improvement or requires sensitive hyperparameter tuning. We restate this challenge as the absence of explicit training signals for the value function in parts of the state–action space. A more effective approach is to provide explicit training signals across the entire action space to eliminate overestimation. We introduce a surprisingly simple yet effective method: **Action-Space Pseudo-Labeling (ASPL)** to resolve this challenge. It completes the value-function's missing signals by assigning pseudo Q-targets that decrease with distance from the behavior support (i.e., the support of the behavior policy). In practice, ASPL achieves an implicit behavior-aware regularization that strengthens as behavior likelihood decreases. On D4RL datasets, we observe stable training and consistent improvements over strong offline baselines with minor tuning burden. Code for reproducing the experiments is provided in the supplementary material.

## 1 INTRODUCTION

Offline reinforcement learning (Fujimoto et al., 2019; Levine et al., 2020) defines a paradigm that learns a policy from a fixed dataset collected by arbitrary, often unrevealed behavior policies. Compared with traditional reinforcement learning, offline RL can be more efficient and safer, as online interaction during training is costly and potentially dangerous. However, the lack of online interaction also makes offline RL challenging. The central difficulty is estimation error caused by distributional shift (Levine et al., 2020), the discrepancy between the learned policy and the behavior policy that collected the dataset. During training, the learned policy is optimized to surpass the behavior policy, which requires estimating the value of out-of-distribution (OOD) actions. Without online feedback, value extrapolation for OOD actions is unreliable, biasing the policy toward overestimated actions and degrading performance.

In this paper, we introduce a new perspective on the analysis of distributional shift issue in offline RL by connecting offline RL with semi-supervised learning (SSL). In SSL, models must estimate labels for samples lacking explicit supervision because only part of the data carries ground-truth signals (Zhu, 2005). By analogy, offline RL must estimate values for out-of-distribution (OOD) actions since datasets cover only a subset of the state–action space. This challenge is structurally analogous: both settings require principled estimation where explicit training signals are absent. In SSL, to address this challenge, a widely used method is pseudo-labeling (Lee et al., 2013): generate labels for unlabeled data and retrain on the union of labeled and pseudo-labeled examples. Inspired by this connection, we propose **Action-Space Pseudo-Labeling (ASPL)**, a minimal extension to a standard actor–critic training that augments training solely with pseudo-labeling actions randomly sampled from the entire action space.

In practice, ASPL jointly trains the Q-network (value function approximator) (Sutton & Barto, 2018) on behavior data and sampled state–action pairs with pseudo Q-targets decaying with distance to behavior support. During each Q-network update, for every state in the minibatch, we randomly sample several actions from the entire action space to form state–action pairs and assign pseudo Q-targets. We then mix these pseudo-labeled state-action pairs with behavior data for joint training,

exposing the Q-network to signals across the entire action space rather than only behavior-supported actions. Accordingly, the pseudo Q-targets do not aim to estimate true action values but instead provide informative training signals when ground-truth targets are absent. In particular, the pseudo Q-target decreases with the distance between the sampled pair and the behavior support. Actions farther from the behavior support are assigned lower targets. This imposes implicit regularization on policy improvement and mitigates overestimation for out-of-distribution actions.

Prior works have also explored leveraging distance-based regularization to mitigate extrapolation bias. Regularization is commonly incorporated in two ways: as a value penalty in the Q-function update and as policy regularization during policy improvement. When estimating Q-values for actions in the dataset, value function regularization adds a distance-based penalty to the standard Bellman target, thereby discouraging overestimation (Kostrikov et al., 2021a; Mao et al., 2023; Huang et al., 2024). On the policy side, policy constraint methods enforce explicit distance-based constraints to the behavior policy during policy improvement from the learned Q-function (Wu et al., 2019; Kumar et al., 2019; Fujimoto & Gu, 2021; Tarasov et al., 2023; Mao et al.).

Despite these efforts, supervision covers only a subset of the action space, leaving the learned value functions unreliable on OOD actions. This forces delicate weight tuning to avoid either over-regularizing or propagating biased estimation. To address this issue, we thus take a different route: complete the missing training signal at the Q-function update by assigning distance-aware pseudo Q-targets ($\tilde{Q}$) to randomly sampled actions, therefore value learning no longer hinges on extrapolation in unsupported regions. Concretely, the induced critic update yields an **implicit behavior-aware weight** that downweights the pseudo Q-target where the behavior policy has mass and strengthens it where coverage is scarce. This reduces to a simple gate—Bellman-dominated on the behavior support and pseudo Q-target–dominated off the behavior support (see Eq. 9). ASPL is plug-and-play for actor–critic pipelines, adds no actor-side constraints, and markedly reduces tuning burden. In extensive D4RL experiments, we observe stable training and consistent gains over strong offline baselines. The analysis and algorithmic details are in Sec. 4 and the empirical study are in Sec. 5.

## 2 PRELIMINARIES

**Reinforcement Learning**. Reinforcement learning (Sutton & Barto, 2018) is defined in the context of Markov Decision Process (MDP) (Puterman, 1990) represented by a tuple $(\mathcal{S}, \mathcal{A}, T, r, \gamma)$, where $\mathcal{S}, \mathcal{A}$ represent space of states and actions respectively, $T(s'|s, a)$ represents environment dynamics, $r(s, a)$ represents reward function, and $\gamma$ represents discount factor. An Agent decides its action by a policy function, $\pi(a|s)$, mapping from states to actions. State visitation frequency induced by $\pi$ is represented by $d^\pi(s)$. The goal of RL is to obtain the optimal policy that maximizes the expectation of return, namely cumulative discounted rewards, which is measured by Q-function. The Q-function following policy $\pi(a|s)$ is defined as:

$$Q(s, a) = \mathbb{E}_{\substack{s_{t+1} \sim T(\cdot|s_t, a_t) \\ a_t \sim \pi(\cdot|s)}} \left[ \sum_{t=0}^{\infty} \gamma^t r(s_t, a_t) \right]. \tag{1}$$

**Offline RL**. In offline RL, datasets are static and previously-collected. An offline Dataset is defined as a set of transitions: $\mathcal{D} = \{(s, a, s', r)\}$. We assume that offline datasets are sampled by an unknown behavior policy $\pi_\beta$. Offline RL based on approximate dynamic programming are typically based on actor-critic framework, which alternates between policy evaluation and policy improvement. In the policy evaluation phase, $Q^\pi$ is calculated via Bellman backup (Sutton & Barto, 2018): $\mathcal{B}^\pi Q(s, a) = r(s, a) + \gamma \mathbb{E}_{s' \sim T(\cdot|s, a) a' \sim \pi(\cdot|s)} [Q(s', a')]$, where $\mathcal{B}^\pi$ is the Bellman operator following $\pi$. The objective for policy improvement is to maximize expected Q-value. In the $k$-th update iteration, the objective for critic and actor training are:

$$\text{Policy evaluation (critic):} \quad \hat{Q}^{k+1} \leftarrow \arg\min_Q \mathbb{E}_{(s,a) \sim \mathcal{D}} \left[ Q(s, a) - \mathcal{B}^{\hat{\pi}^k} \hat{Q}^k(s, a) \right]^2 \tag{2}$$

$$\text{Policy improvement (actor):} \quad \hat{\pi}^{k+1} \leftarrow \arg\max_\pi \mathbb{E}_{\substack{s \sim \mathcal{D} \\ a \sim \pi(\cdot|s)}} \left[ \hat{Q}^k(s, a) \right] \tag{3}$$

Here $\hat{Q}$ and $\hat{\pi}$ denote an approximator of $Q$ and $\pi$ respectively, which are usually a neural network in practice.

## 3 RELATED WORKS

**Distance-based regularization.** A popular methods in prior works is mitigating extrapolation bias by penalizing the distance of actions from the behavior support or their divergence from the behavior policy, instantiated either on the critic side or the actor side. **Critic-side** value regularization augments the Bellman target with a distance penalty that discourages large values on actions far from the behavior support; a representative form is

$$\hat{Q}^{k+1} \leftarrow \arg\min_Q \mathbb{E}_{(s,a)\sim\mathcal{D}}[Q(s,a) - (\mathcal{B}^{\hat{\pi}}\hat{Q}^k(s,a) - \alpha \cdot d(\hat{\pi}^k, \pi_\beta))]^2, \qquad (4)$$

where $d(\hat{\pi}, \pi_\beta)$ measures the action-space distance from $a$ to the support of the behavior policy at state $s$ (e.g., supported value regularization (Mao et al., 2023; Huang et al., 2024)); Fisher-divergence critic regularization penalizes gradients of $Q$ to smooth values off-support (Kostrikov et al., 2021a). **Actor-side** policy regularization constrains policy improvement to remain close to the behavior policy via a distance measurement $d(\hat{\pi}, \pi_\beta)$ (e.g., KL/MMD/Wasserstein or $L_2$ behavior regression) (Wu et al., 2019; Kumar et al., 2019; Fujimoto & Gu, 2021; Tarasov et al., 2023), with a typical objective

$$\hat{\pi}^{k+1} \leftarrow \arg\max_\pi \mathbb{E}_{s\sim\mathcal{D}, a\sim\pi(\cdot|s)}[\hat{Q}^k(s,a) - \alpha \cdot d(\pi, \pi_\beta)]. \qquad (5)$$

or equivalently through weighted maximum-likelihood updates that implicitly realize the constraint (Nair et al., 2020; Wang et al., 2020; Peng et al., 2019). Both families reduce overestimation in unsupported regions by discouraging OOD actions, but rely on values learned from partial coverage of the action space and often require careful tuning of the regularization coefficient.

Beyond distance-based regularization, we highlight several representative offline actor–critic methods widely used in practice. Conservative Q-Learning (CQL) learns a $Q$-function that provably lower-bounds the true value of a policy by augmenting the Bellman objective with a conservative (e.g., log-sum-exp) regularizer, thereby producing pessimistic estimates for unseen actions while retaining standard Q-learning (Kumar et al., 2020; Lyu et al., 2024). Implicit Q-Learning (IQL) avoids explicit behavior constraints by estimating state values via upper expectile regression on in-dataset actions and extracting the policy with advantage-weighted regression, enabling in-sample learning without querying unseen actions (Kostrikov et al., 2021b; Xu et al., 2023; Chen et al.). Uncertainty-based methods leverage epistemic uncertainty—typically via ensembles or variance proxies—to downweight high-uncertainty regions during value learning or policy improvement, yielding more conservative estimates and safer updates (An et al., 2021; Wang et al.; Bai et al., 2022).

## 4 ACTION-SPACE PSEUDO-LABELING

To tackle the core challenge of offline RL: sparse coverage in the action space leaves the critic without supervision beyond the behavior support, inducing overestimation on OOD actions, we present **Action-Space Pseudo-Labeling (ASPL)** as a critic-side augmentation that assigns distance-aware pseudo Q-targets to randomly sampled actions, thereby extending supervision across the entire action space while imposing no actor-side constraints and reducing hyperparameter sensitivity. Sec. 4.1 states the objective for policy evaluation and specifies the construction of pseudo Q-targets; Sec. 4.2 analyzes the induced behavior-aware weighting that emphasizes the Bellman target on the behavior support and the pseudo Q-target where coverage is limited. Sec. 4.3 details implementation choices and integration into standard actor–critic pipelines.

### 4.1 PSEUDO-LABELING IN POLICY EVALUATION

The absence of explicit training signals for OOD actions leading to overestimation, harming performance. We aim to address this issue by introducing a minimal addition to a standard actor–critic training: a critic-side pseudo-label regression term for randomly sampled actions. This modification adds only minor overhead to the critic update compared with a standard actor–critic. During critic training, we form random state–action pairs by combining dataset states with actions randomly sampled from the entire action space. Then we mix them with state–action pairs sampled from the dataset for joint optimization. For dataset state–action pairs we apply the standard Bellman backup;

for random state–action pairs we regress to pseudo Q-targets, thereby providing explicit supervision for the entire action space. The overall critic update is represented as

$$\hat{Q}^{k+1} \leftarrow \arg\min_{Q} \mathbb{E}_{(s,a)\sim\mathcal{D}}[Q(s,a) - \mathcal{B}^{\pi^k}\hat{Q}^k(s,a)]^2 + \alpha \mathbb{E}_{\substack{s\sim\mathcal{D}\\ \tilde{a}\sim\mathcal{U}}}[Q(s,a) - \tilde{Q}(s,\tilde{a})]^2, \quad (6)$$

Here we denote by $\mathcal{U}$ an action distribution that entirely covers the action space. As a result, we modify only the critic loss, introduce no new training phases or additional neural networks, and add no policy-improvement regularization. This design interleaves in-dataset state-action pairs with randomly sampled candidates to expand coverage beyond the behavior support while preserving data-anchored targets, thereby stabilizing critic learning.

A critical design in our method is assigning distance-based pseudo labels, namely the pseudo Q-targets, to the random state–action pairs. Because environment interaction is unavailable, these random state–action pairs lack ground-truth rewards. We therefore use pseudo-labeling to bias the policy toward in-distribution actions rather than evaluating the ground-truth Q-value. Specifically, pseudo-labels impose a stronger penalty on actions farther from the behavior support. In practice, for a state–action pair $(s,a)$ in the $k$-th minibatch, we sample $N$ random actions, represented as $\{\tilde{a}\}$, and their pseudo-labels are computed as:

$$\tilde{Q}(s,\tilde{a}) = \hat{Q}^k(s,a) - d(a,\tilde{a}) \quad (7)$$

where $\hat{Q}^k(s,a)$ serves as the in-distribution anchor and $d(a,\tilde{a}) \geq 0$ is a distance measurement that decreases monotonically with action-space distance. During critic training, $\tilde{Q}$ is treated as a stop-gradient target. Since the pseudo-label value decreases monotonically with distance via $d(\cdot)$, random actions farther from the behavior support receive lower pseudo-labels. Consequently, the critic learns value estimates that suppress Q-values for unsupported actions, guiding policy to remain aligned with the behavior while maximizing $\hat{Q}$.

Intuitively, pseudo-labels provide a conservative teaching signal for OOD action regions, preventing unsupported inflation of the state–action value function and thereby mitigating overestimation risk. The overall critic update can be equivalently interpreted as a weighted mixture of the standard Bellman backup target and a pseudo-label target. The Bellman backup component acting as a reliable anchor for in-distribution actions so that estimates remain stable on the behavior support. Complementarily, the pseudo-label term is applied only to randomly sampled actions and linearly attenuates their assigned values with increasing action-space distance from the dataset support, which supplies supervision where the data are sparse and suppresses OOD overestimation. This augmentation leaves the actor update and the training-time complexity unchanged, making the method a plug-and-play component for existing actor–critic pipelines.

## 4.2 BEHAVIOR-AWARE WEIGHT

With a joint critic loss, the central question is how to balance its two terms—the Bellman-target term and the pseudo-label term. We argue that ASPL implicitly achieves this balance: it assigns more weight to the Bellman target in regions with high coverage under the behavior policy $\pi_\beta(a|s)$ and more weight to the pseudo-label in regions with low coverage, thereby mitigating extrapolation error. We formalize this intuition by analyzing the critic update in Eq.( 6). Let $u(a)$ denotethe density of $\mathcal{U}$. By greedily solving the optimization problem in Eq. (6), $\hat{Q}^{k+1}$ can be expressed in terms of $Q^k$ as

$$\hat{Q}^{k+1}(s,a) = (1 - w_\beta(s,a))\mathcal{B}^{\hat{\pi}^k}Q^k(s,a) + w_\beta(s,a)\tilde{Q}(s,a), \quad (8)$$

with the weight of the pseudo label term is calculated by

$$w_\beta(s,a) = \frac{\alpha\, u(a)}{\pi_\beta(a|s) + \alpha\, u(a)}. \quad (9)$$

When $\mathcal{U}$ is uniform and $u(a)$ is constant, the intuition follows directly: in high-coverage regions ($\pi_\beta(a|s) \gg \alpha\, u(a)$), $w_\beta(s,a)$ is small and the update is Bellman-dominated, whereas in low-coverage or OOD regions ($\pi_\beta(a|s) \rightarrow 0$), $w_\beta(s,a) \rightarrow 1$ and the update follows the pseudo-label, yielding conservative distance-based estimates outside the dataset support while maintaining fidelity within the dataset support.

In fact, building on Eq. (8), the behavior-aware weight decreases with behavior coverage whenever the sampling density satisfies

$$\frac{\partial \ln u(a)}{\partial \ln \pi_\beta(a|s)} < 1. \tag{10}$$

Under the above condition, $w_\beta(s, a)$ is strictly decreasing in $\pi_\beta(a|s)$. Hence for any fixed $s$ and actions $a_1, a_2$ with $\pi_\beta(a_1|s) > \pi_\beta(a_2|s)$, we have $w_\beta(s, a_1) < w_\beta(s, a_2)$, so the update allocates relatively more pseudo-label weight to the lower-coverage action. This yields a smooth, threshold-free reweighting rule that steers the critic toward Bellman targets where coverage is high and toward Pseudo Q-targets where coverage is low. The detailed derivation can be found in Appendix B.

**Multi-Arm Bandits.** To explain the effect of behavior-aware weight, we compare ASPL to TD3+BC in a one-dimensional continuous bandit on $[-1, 1]$ with a multimodal behavior policy. TD3+BC performs policy improvement via Eq. (5) with $d(\hat{\pi}, \pi_\beta) = \mathbb{E}_{a \sim \hat{\pi}(\cdot|s), \tilde{a} \sim \pi_\beta(\cdot|s)}[(a - \tilde{a})^2|s]$ (Fujimoto & Gu, 2021). Fig. 1 visualizes the policy-improvement landscape for each method after 2000 gradient steps. As shown in the result, ASPL mitigates overestimation off-support and is comparatively **insensitive** to $\alpha$; for sufficiently small $\alpha$, the learned $Q$ closely tracks the true $Q$ near the dataset-supported optimum. By contrast, TD3+BC tends to overestimate for small $\alpha$, while large $\alpha$ over-regularizes the policy, highlighting tuning sensitivity in multimodal settings.

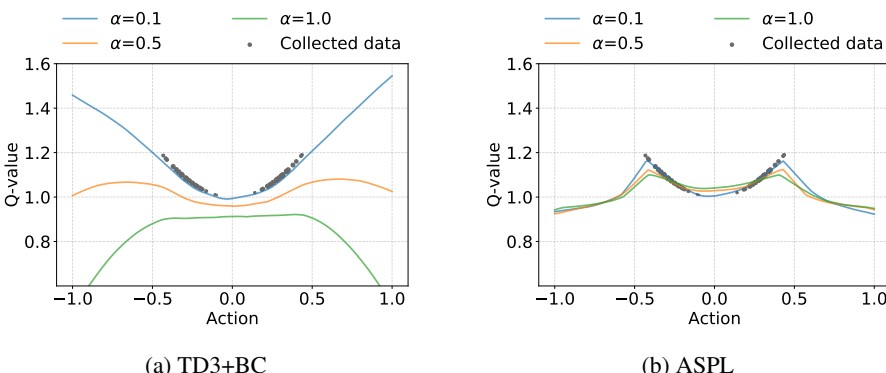

(a) TD3+BC                                     (b) ASPL

Figure 1: The landscape of policy improvement objectives under different regularization weights (2000 gradient steps). Scatter size is proportional to the behavior-policy selection probability. TD3+BC exhibits strong dependence on the regularization scale (small: overestimation; large: over-regularization). ASPL's behavior-aware weighting mitigates overestimation across a range of $\alpha$; for sufficiently small $\alpha$, the objective closely approximates the $Q$-function within the behavior support, while assigning to OOD actions pseudo Q-targets decaying with the distance to the behavior support.

### 4.3 IMPLEMENTATION DETAILS

We implement ASPL by adding a lightweight addition to a standard actor–critic training loop: the critic objective is augmented with a pseudo-label regression term (Eq. 6), where pseudo-labels are defined by Eq. 7; and the actor update is left unchanged. In practice, we build our algorithm on top of TD3 (Fujimoto et al., 2018), one of the existing state-of-the-art online RL algorithms. Actions sampled from $\mathcal{U}$ for the pseudo-label term are drawn with Latin Hypercube Sampling (LHS) over the bounded action space. LHS yields uniform marginals along every dimension and, in practice, an approximately constant joint density inside the bounded hyper-rectangle (and zero outside), consistent with the analysis that assumes a uniform coverage distribution $\mathcal{U}$.

To align the pseudo-label penalty with the value scale and make distances comparable across dimensions, we define the distance term as a range-normalized squared difference multiplied by a global scale:

$$d(a, \tilde{a}) = c \cdot \left(\frac{a - \tilde{a}}{a_{\max} - a_{\min}}\right)^2. \tag{11}$$

The scale $c$ is the running average of $|\hat{Q}(s,a)|$ over all state–action pairs sampled from the dataset during training up to the current iteration. Intuitively, the range-normalized squared distance provides a dimensionless measure of separation in action space, while the global scale keeps the penalty on the same numerical order as the evolving $Q$-values. As in Eq. 7, the pseudo-label $\tilde{Q}$ is used as a stop-gradient target.

---

**Algorithm 1:** Actor-Space Pseudo-Labeling

1   Initialize a Q-network $\hat{Q}$ and a policy network $\hat{\pi}$.
2   **for** *each training step* **do**
3      Sample a batch $\{s, a, s', a', r\}$ from the dataset.
4      Sample $N$ random action $\tilde{a}$ from $\mathcal{U}$
5      Calculate $\tilde{Q}(s,a)$ for $(s, \tilde{a})$ by Eq.(7)
6      Update $\hat{Q}$ by Eq.( 6)
7      Update $\hat{\pi}$ by Eq.( 3)
8   **end**

---

**In summary**, ASPL adopts a distinct approach defined by Eq. (6): instead of penalizing values (critic-side regularization) or constraining the policy (actor-side regularization), it directly supervises actions drawn from $\mathcal{U}$ by regressing distance-based pseudo-labels. Building on the analysis in Sec. 4.2, the joint objective induces an **implicit behavior-aware weight** that trades off the Bellman target and the pseudo-label target. As shown by Eq. (8), this weight decreases with behavior coverage: updates are Bellman-dominated on-support actions and pseudo-label–dominated off-support. Practically, ASPL is plug-and-play for standard actor–critic pipelines: it modifies only the critic loss, introduces no actor-side constraints or auxiliary networks, and leaves training-time complexity essentially unchanged. The behavior-aware weighting mitigates sensitivity to regularization scales, thereby reducing hyperparameter tuning burden while preserving stability under sparse coverage. Empirically (Sec. 5), ASPL achieves competitive or superior normalized scores on D4RL Gym–MuJoCo tasks and exhibits early convergence together with robustness to the number of sampled actions and the weighting coefficient, offering a simple yet effective baseline for offline RL.

## 5   EXPERIMENTAL EVALUATION

To assess the effectiveness of ASPL, we conduct a comprehensive set of experiments on the D4RL (Fu et al., 2020) benchmark, focusing on widely used Gym–MuJoCo locomotion tasks. Our evaluation aims to determine whether ASPL achieves stable policy improvement and competitive performance across diverse offline datasets, in comparison with representative offline RL baselines including TD3+BC, CQL, IQL, OAC–BVR, and SCAS. Beyond these benchmark comparisons, we further examine the sensitivity of ASPL to key hyperparameters, thereby providing a more complete picture of its robustness and practical applicability.

### 5.1   EVALUATION ON OFFLINE BENCHMARKS

**benchmark.** We evaluate on the D4RL (Fu et al., 2020) Gym–MuJoCo locomotion (Brockman et al., 2016) tasks—hopper, walker2d and halfCheetah–using the official normalized scoring protocol, where a score of $0$ corresponds to the return of a uniformly random policy and $100$ corresponds to a domain-specific expert for each environment. Unless otherwise specified, we use the official "$-\mathtt{v2}$" datasets and report results averaged over four random seeds for comparability. In this section, all reported scores are computed as the average over the final 10 evaluations conducted during training.

Within this Gym domain, our experiments cover the following dataset types: **medium** (trajectories collected by a policy trained online and early-stopped at an intermediate performance level, yielding suboptimal yet structured behavior), **medium-replay** (the entire replay buffer recorded during training of the medium-level agent up to the medium performance point, providing a heterogeneous mixture of on- and off-policy samples with broader state–action coverage), and **medium-expert** (an equal-proportion mixture of medium and expert trajectories that forms a multi-modal dataset probing

Table 1: Normalized score of ASPL and baselines on Gym domains from D4RL, averaged over four seeds. The scores that exceed 0.95x the best one are highlighted.

| Task Name | TD3BC[†] | CQL[†] | IQL[†] | OAC-BVR | SCAS | ASPL |
|---|---|---|---|---|---|---|
| hopper-medium | 60.10 | 55.45 | 65.70 | 95.00 | 102.50 | 80.06 |
| walker2d-medium | 80.70 | 74.40 | 79.50 | 86.00 | 82.30 | 85.90 |
| halfcheetah-medium | 51.50 | 44.95 | 48.70 | 52.20 | 46.60 | 57.64 |
| hopper-medium-replay | 76.55 | 93.95 | 90.55 | 95.30 | 109.70 | 95.57 |
| walker2d-medium-replay | 89.05 | 90.35 | 92.50 | 77.30 | 108.40 | 80.77 |
| halfcheetah-medium-replay | 67.85 | 58.45 | 64.40 | 48.30 | 91.70 | 52.05 |
| hopper-medium-expert | 85.25 | 95.60 | 101.75 | 96.50 | 101.60 | 109.17 |
| walker2d-medium-expert | 96.10 | 93.45 | 92.95 | 112.00 | 78.10 | 109.77 |
| halfcheetah-medium-expert | 69.00 | 69.40 | 69.85 | 93.10 | 44.00 | 95.41 |
| Average | 75.12 | 75.11 | 78.43 | 83.97 | 84.99 | 85.15 |

an algorithm's ability to learn from mixed-quality experience). We follow the D4RL evaluation protocol and report normalized scores for the above tasks and dataset types to enable apples-to-apples comparisons across methods.

**Baselines.** On the D4RL benchmark (Fu et al., 2020), we compare ASPL against five algorithms. Specifically, we include three classical offline RL methods: (1) TD3+BC (Fujimoto & Gu, 2021), which augments TD3 with a behavior-cloning regularizer in the actor objective to balance value maximization and imitation under offline data; (2) IQL (Kostrikov et al., 2021b), which learns value functions via expectile regression and performs advantage-weighted behavior cloning to avoid explicit behavior-policy constraints; (3) CQL (Kumar et al., 2020), which penalizes Q-values on out-of-distribution actions to obtain conservative critics that mitigate extrapolation error and two recent methods without training auxiliary networks beyond the actor and critic: (4) OAC–BVR (Huang et al., 2024), and (5) SCAS (Mao et al.), For fair comparison, we re-ran TD3+BC[1], IQL[2], and CQL[3] using the author-provided implementations on the -v2 environments. For OAC–BVR and SCAS, we report the scores provided in the original papers.

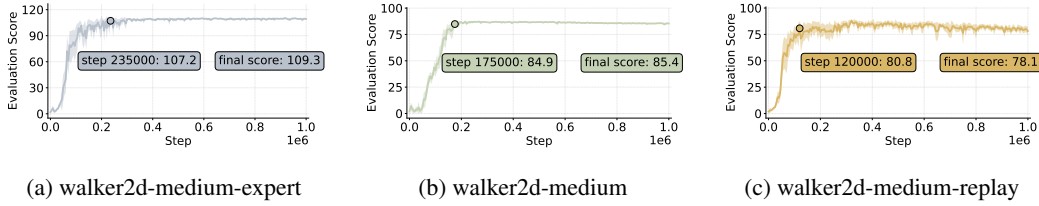

(a) walker2d-medium-expert  (b) walker2d-medium  (c) walker2d-medium-replay

Figure 2: We evaluate walker2d training as follows: every 5,000 training steps we run 10 evaluation episodes and report the mean episodic return. The solid curve plots the average across 12 random seeds, and the shaded band denotes the 95% confidence interval. Dots mark, for each method, denote the first checkpoint at which the average score exceeds 98% of its final average score. ASPL exhibits stable learning dynamics: it reaches near–final performance after comparatively few training steps and maintains that level for the remainder of training.

**Results.** Table 1 reports evaluation on the D4RL benchmark and shows that ASPL delivers the highest average normalized score across the nine D4RL Gym–MuJoCo tasks. We highlight that all ASPL results are obtained under a **fixed hyperparameter setting** ($\alpha = 0.05$, $N = 6$). It achieves three clear wins (halfcheetah-medium, hopper-medium-expert, halfcheetah-medium-expert) and is within 5% of the best on walker2d-medium-expert and walker2d-medium. Figure 4 shows the training process on walker2d datasets: the learned policy improves steadily, reaches competitive test returns

---

[1]https://github.com/sfujim/TD3_BC

[2]https://github.com/ikostrikov/implicit_q_learning

[3]https://github.com/aviralkumar2907/CQL

after few training iterations, and remains stable thereafter. Overall, the combination of average normalized score and stable training dynamics scores ASPL as a strong offline RL baseline.

## 5.2 Hyperparameter Studies

We evaluate ASPL under multiple hyperparameter settings across four random seeds and use $\alpha = 0.15$, $N = 6$ as the default. Varying $N$ shows that ASPL is remarkably insensitive to the sampling budget: even with $N = 1$ (a single random action), performance remains strong and close to the default. This observation alleviates concerns about the effectiveness of action sampling in high-dimensional spaces, suggesting that ASPL's pseudo-labeling objective does not rely on exhaustive candidate enumeration.

For the weighting coefficient $\alpha$, smaller values consistently yield better results, consistent with the behavior-aware weighting in Eq. (8): the pseudo-label weight increases with $\alpha$ and decreases with behavior coverage, so reducing $\alpha$ uniformly down-weights the pseudo-label term and shifts updates toward the Bellman target, mitigating excessive pessimism while retaining off-support regularization via randomly sampled pairs.

Overall, our experiments indicate that ASPL attains competitive performance with minimal tuning: it is robust to the choice of the number of sampled actions and achieves its best performance under mildly conservative $\alpha$. Through the behavior-aware weight, such settings emphasize the Bellman target in high-coverage regions while retaining conservative pseudo-label guidance in low-coverage regions.

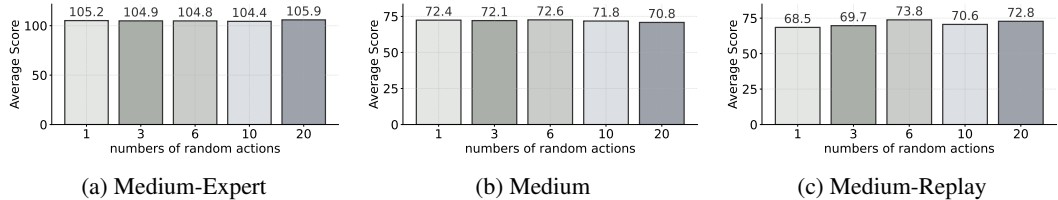

(a) Medium-Expert  (b) Medium  (c) Medium-Replay

Figure 3: Effect of the number $N$ of randomly sampled actions per state for pseudo-labeling. Average normalized D4RL scores over four seeds on the medium-expert, medium, and medium-replay datasets. Results show ASPL is robust to the sampling budget: performance remains competitive even at $N = 1$, indicating exhaustive action enumeration is unnecessary for stable gains.

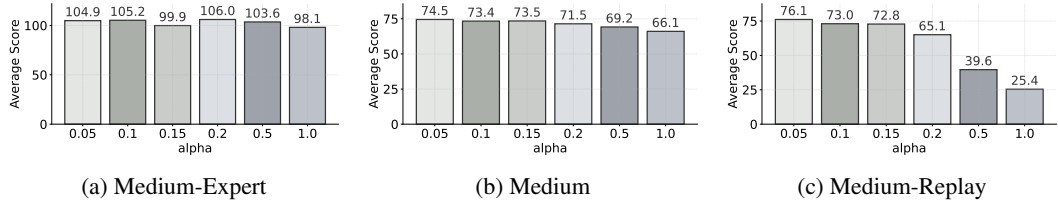

(a) Medium-Expert  (b) Medium  (c) Medium-Replay

Figure 4: Sensitivity of ASPL to the pseudo-label weight $\alpha$. Average normalized D4RL scores over four seeds on the medium-expert, medium, and medium-replay datasets as $\alpha$ varies; smaller $\alpha$ yields higher returns by emphasizing Bellman targets in high-coverage regions while retaining conservative off-support guidance.

## 6 Conclusion

In this work we revisited the challenge of out-of-distribution action evaluation in offline reinforcement learning and introduced Action-Space Pseudo-Labeling (ASPL), a lightweight critic-side mechanism that supplements supervision over the entire action space by assigning pseudo-targets that decay monotonically with distance from the behavior support. By augmenting the temporal-difference objective with this coverage-aware signal—without imposing explicit policy constraints or adding auxiliary networks—ASPL yields a simple yet effective bias toward behavior-supported

regions while still enabling value learning where logged supervision is sparse. Empirically, a TD3-based instantiation of ASPL delivers stable training dynamics and competitive normalized returns across D4RL Gym–MuJoCo benchmarks, with reduced sensitivity to sampling budgets and weighting coefficients. Looking forward, we see several promising directions: formal analyses of convergence and approximation error induced by pseudo-labeling; adaptive, state-dependent weighting and uncertainty-aware proposal distributions; richer targets and learned distance metrics; integration with conservative baselines, ensemble critics, and model-based planning; extensions to discrete or mixed action spaces, partial observability, multi-task and multi-agent settings; and applications in safety-critical domains where behavior support estimation and constraint handling are central. Collectively, these avenues position ASPL as a general principle for action-space supervision completion in offline decision making.

## ETHICS STATEMENT

This study uses only publicly available offline RL benchmarks in simulated environments; no new data were collected and no personally identifiable information, human subjects, or animal experiments are involved. There is no online interaction or real-world deployment, so risks to privacy and safety are minimal. While any data-driven method may inherit dataset biases, our evaluations are confined to non-human control tasks. Compute requirements are moderate, and we will release code and configurations for transparent assessment.

## REPRODUCIBILITY STATEMENT

All hyperparameters and experimental configurations are reported in the Appendix C to ensure transparency and reproducibility.

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

## A  USE OF LLMS

**LLM Usage Statement.**  We used a large language model (LLM) as a general-purpose assistant for grammar editing and minor wording suggestions. The model did not generate novel scientific claims, analyses, or experimental results. All technical content (problem formulation, theorems, proofs, algorithm design, and experiments) was authored and verified by the listed authors. We manually reviewed and fact-checked all LLM-suggested edits and take full responsibility for the final manuscript.

## B  PROOF

### B.1  Q-FUNCTION ITERATION

We restate the training objective:

$$\hat{Q}^{k+1} \leftarrow \arg\min_{Q} \; \mathbb{E}_{(s,a)\sim\mathcal{D}}\big[Q(s,a) - \mathcal{B}^{\pi^k}\hat{Q}^k(s,a)\big]^2 + \alpha\, \mathbb{E}_{\substack{s\sim\mathcal{D}\\a\sim\mathcal{U}}}\big[Q(s,a) - \tilde{Q}(s,a)\big]^2, \quad (12)$$

**Pointwise minimization.**  Fix $(s,a)$. The population objective induced by equation 12 reduces (up to a positive multiplicative constant) to a weighted least squares in the single scalar $Q(s,a)$:

$$\ell_{s,a}(Q) \;=\; \pi_\beta(a\,|\,s)\big(Q(s,a) - \mathcal{B}^{\pi^k}\hat{Q}^k(s,a)\big)^2 \;+\; \alpha\, u(a)\big(Q(s,a) - \tilde{Q}(s,a)\big)^2,$$

which is strictly convex in $Q(s,a)$. Setting the derivative to zero gives

$$\frac{\partial \ell_{s,a}}{\partial Q(s,a)} = 2\,\pi_\beta(a\,|\,s)\Big(Q(s,a) - \mathcal{B}^{\pi^k}\hat{Q}^k(s,a)\Big) + 2\,\alpha\,u(a)\Big(Q(s,a) - \tilde{Q}(s,a)\Big) = 0.$$

Solving for $Q(s,a)$ yields the unique minimizer

$$\begin{aligned}
Q^\star(s,a) &= \frac{\pi_\beta(a\,|\,s)}{\pi_\beta(a\,|\,s) + \alpha\,u(a)}\,\mathcal{B}^{\pi^k}\hat{Q}^k(s,a) + \frac{\alpha\,u(a)}{\pi_\beta(a\,|\,s) + \alpha\,u(a)}\,\tilde{Q}(s,a) \\
&= \big(1 - w_\beta(s,a)\big)\mathcal{B}^{\pi^k}\hat{Q}^k(s,a) + w_\beta(s,a)\,\tilde{Q}(s,a),
\end{aligned}$$

with

$$w_\beta(s,a) = \frac{\alpha\,u(a)}{\pi_\beta(a\,|\,s) + \alpha\,u(a)}.$$

Assigning the minimizer to the next iterate recovers the paper's updates:

$$\hat{Q}^{k+1}(s,a) = (1 - w_\beta(s,a))\,\mathcal{B}^{\pi^k}\hat{Q}^k(s,a) + w_\beta(s,a)\,\tilde{Q}(s,a), \quad (8)$$

$$w_\beta(s,a) = \frac{\alpha\,u(a)}{\pi_\beta(a\,|\,s) + \alpha\,u(a)}. \quad (9)$$

### B.2  MONOTONICITY OF THE BEHAVIOR-AWARE WEIGHT

Recall

$$w_\beta(s,a) = \frac{\alpha\,u(a)}{\pi_\beta(a\,|\,s) + \alpha\,u(a)},$$

and allow the sampling density $u(a)$ to (weakly) depend on $\pi_\beta(a\,|\,s)$ so that $u = u\big(\pi_\beta(a\,|\,s)\big)$. For brevity, write $\pi_\beta \equiv \pi_\beta(a\,|\,s)$ and $u' \equiv \frac{\partial u}{\partial \pi_\beta}$. Differentiating $w_\beta$ with respect to $\pi_\beta$ yields

$$\frac{\partial w_\beta}{\partial \pi_\beta} = \frac{\alpha\big(\pi_\beta u' - u\big)}{\big(\pi_\beta + \alpha u\big)^2}.$$

Since the denominator is strictly positive, we have

$$\frac{\partial w_\beta}{\partial \pi_\beta} < 0 \quad\Longleftrightarrow\quad \pi_\beta\, u' - u < 0 \quad\Longleftrightarrow\quad \frac{1}{u}\frac{\partial u}{\partial \pi_\beta} < \frac{1}{\pi_\beta} \quad\Longleftrightarrow\quad \frac{\partial \ln u(a)}{\partial \ln \pi_\beta(a\,|\,s)} < 1. \quad (10)$$

**Special case (uniform sampler).** If $\mathcal{U}$ is uniform, then $u$ is constant and $u' = 0$, so

$$\frac{\partial w_\beta}{\partial \pi_\beta} = \frac{-\alpha\, u}{\left(\pi_\beta + \alpha u\right)^2} \; < \; 0,$$

which verifies monotone decrease without additional assumptions.

## C EXPERIMENT DETAILS

All experiments were conducted under a standardized software stack to ensure reproducibility. The operating system was Linux (kernel 6.14.0-29-generic) with Python 3.11.13 as the main programming environment. Core libraries included PyTorch 2.5.1 (compiled with CUDA 12.1 and cuDNN 9.1), Gym 0.23.1, D4RL 1.1, and MuJoCo 3.3.3 interfaced via mujoco-py 2.1.2.14. For hardware, training and evaluation were performed on a single NVIDIA GeForce RTX 4080s GPU. No multi-GPU or distributed training was used in any reported results. All parameters in the experiment are listed here.

Table 2: Training and evaluation hyperparameters used in all experiments unless otherwise noted. Only parameters not already specified in the main text are listed.

| Parameter Description | Value |
|---|---|
| Maximum training timesteps | 1,000,000 |
| Evaluation frequency | 5,000 |
| Episodes per evaluation during training | 10 |
| Hidden layer dimension (MLP) | 256 |
| Number of hidden layers (actor MLP) | 3 |
| Number of hidden layers (critic MLP) | 4 |
| Optimizer learning rate | $3 \times 10^{-4}$ |
| Batch size | 512 |
| Discount factor $\gamma$ | 0.99 |
| Target network update rate $\tau$ | 0.005 |
| Target policy noise (std) for TD3 targets | 0.2 |
| Target noise clip | 0.5 |
| Policy update frequency | 2 |

