# OpenReview forum: "Offline Reinforcement Learning via Action-Space Pseudo-Labeling"
_ICLR.cc/2026/Conference — Submitted to ICLR 2026_

### Official Review · Reviewer_gcNK · 2025-10-28

**Soundness:** 2
**Presentation:** 3
**Contribution:** 2
**Rating:** 2
**Confidence:** 4

**Summary:**

The authors propose Action-Space Pseudo-Labeling (ASPL), a simple method that assigns decreasing pseudo Q-targets to actions based on their distance from the behavior action.

**Strengths:**

Its primary strength lies in proposing a value regularization method, Action-Space Pseudo-Labeling, which directly mitigates overestimation for out-of-distribution actions.

**Weaknesses:**

1. The novelty of the proposed method is limited. The core idea of applying pseudo-labeling to OOD actions is already established in prior works like MCQ, SVR, and OAC-BVR. The paper lacks a compelling justification for why the specific design of ASPL is superior to these existing approaches.
2. There appears to be a potential error in Equation (6), which warrants clarification from the authors.
3. By applying an explicit penalty based on the distance, the proposed method introduces a bias estimation that may also propagate to in-distribution state-action pairs through the Bellman backup. This could systematically depress Q-values across the entire state-action space. The authors should provide Q-value curves for all tasks to investigate this potential issue.
4. The experimental evaluation is insufficient, as it is conducted solely on the Gym domain. For a comprehensive assessment in offline RL, it is essential to include results on more challenging benchmarks, such as the AntMaze domain, which tests an algorithm's ability to perform long-horizon tasks.
5. There is a significant discrepancy in the reported results for the baseline method SCAL on the halfcheetah-medium-expert task (44.0 in this work vs. 91.7 in the original paper). This major inconsistency casts doubt on the integrity of the experimental setup and the validity of the comparative analysis.
6. While the authors cite the most relevant prior works (MCQ and SVR), they conspicuously omit direct comparisons with them. This lack of comparison with state-of-the-art methods that share a similar design philosophy is a critical omission and prevents a fair assessment of the method's contribution.
7.  Given the final performance of ASPL as reported, its overall effectiveness is questionable. A simple comparison with the reported performance of MCQ and SVR would likely confirm this performance gap.
8. The experimental results report only mean performance without the corresponding standard deviations across multiple seeds. Reporting the variance is essential to properly assess the significance and reliability of the results.
9. The use of a batch size of 512 deviates from the community-standard value of 256 used in most prior works for fair comparison. The authors must justify this choice and provide an ablation study.

**Questions:**

Please see Weakness

---

### Official Review · Reviewer_BypT · 2025-10-31

**Soundness:** 2
**Presentation:** 2
**Contribution:** 2
**Rating:** 4
**Confidence:** 4

**Summary:**

The paper proposes Action-Space Pseudo-Labeling (ASPL) for offline RL. ASPL augments standard actor–critic training by sampling additional actions for each state from the entire action space and assigning pseudo Q-targets that decay with distance to the behavior support. These pseudo-labeled state–action pairs are mixed with logged behavior data to train the critic, aiming to mitigate OOD overestimation without introducing explicit policy constraints or auxiliary networks. The implementation is TD3-based; the actor update remains unchanged. The authors report stable training and consistent gains over strong offline baselines on D4RL Gym–MuJoCo.

**Strengths:**

Clear, practical motivation. Frames OOD action value extrapolation as a “missing supervision” issue and fills it via pseudo-labels in the action space.

Low engineering overhead. Only the critic objective is modified; the actor and architecture remain simple—no explicit policy constraints or additional networks.

**Weaknesses:**

Incremental novelty / positioning.
Concept overlaps with conservative value shaping and behavior-regularized approaches. The paper should more rigorously differentiate ASPL from CQL/XQL/IVR-style methods using a clearer formal comparison and targeted ablations.

Baseline scope.
To substantiate broad claims, comparisons to additional SOTA offline RL methods beyond the immediate TD3-family/popular baselines would strengthen the case.

Behavior-support distance estimation.
The core mechanism hinges on how “distance to behavior support” is estimated and scaled. More analysis of bias/variance, dimensionality effects, and sensitivity to the scaling constant is needed.

External validity / harder settings.
Evidence is limited to D4RL Gym–MuJoCo. Including harder OOD regimes (e.g., AntMaze, Adroit, Kitchen, or random datasets) would better validate the central claim in challenging settings.

#Minor Concerns

Figure clarity. The method diagram should explicitly show that pseudo-targets monotonically decay with behavior-distance and how they combine with the Bellman target.

Evaluation protocol upfront. Consolidate seeds, episode counts, and final-N evaluation aggregation in the main text (not only in the appendix).

Compute/resource reporting. Since ASPL samples extra actions, report training time / memory vs. pseudo-sample budget
𝑁

**Questions:**

How is distance to the behavior support estimated in practice, and how sensitive are results to the scaling constant and normalization?

What is the trade-off between pseudo-sample budget 𝑁, wall-clock time, and returns?

Does integrating ASPL with IQL/CQL/XQL critics still help when those methods already reduce OOD overestimation?

How robust is uniform/LHS sampling for high-dimensional actions? Would learned proposal distributions (e.g., uncertainty-aware) improve coverage vs. cost?

Any failure cases when behavior coverage is extremely narrow? How does ASPL behave near the support boundary?

---

### Official Review · Reviewer_tqHU · 2025-10-31

**Soundness:** 2
**Presentation:** 2
**Contribution:** 1
**Rating:** 2
**Confidence:** 3

**Summary:**

To address the overestimation of out-of-distribution (OOD) action values caused by distribution shift in (Offline RL), this paper proposes the Action Space Pseudo-Labeling (ASPL) method. Its core idea draws an analogy to semi-supervised learning: it provides explicit supervision for the entire action space and assigns "distance-aware pseudo-Q targets" to state-action pairs randomly sampled from the entire action space.

**Strengths:**

The paper features a clear logical flow, rigorous expression, and effective experimental visualizations.

**Weaknesses:**

1. The experiments are insufficient, as they only cover Gym environments and lack more complex ones such as Antmaze, maze, and kitchen.
2. The method proposed in the paper falls into the category of conservative Q-values. By subtracting $d(a,\bar{a})$, it shows limited overall innovation and exhibits little difference from CQL.
3. There is a lack of ablation studies on fixed $\omega_{\beta}$ and adaptive weights.

**Questions:**

please refer to weaknesses.

---

### Official Review · Reviewer_iaNB · 2025-10-31

**Soundness:** 2
**Presentation:** 3
**Contribution:** 2
**Rating:** 4
**Confidence:** 2

**Summary:**

This paper addresses the challenge of offline reinforcement learning (offline RL), particularly in overcoming the issue of overestimating values for out-of-distribution (OOD) actions when training from a fixed dataset. It introduces the Action-Space Pseudo-Labeling (ASPL) method, which assigns pseudo Q-values to randomly sampled actions in the state–action space, providing training signals for actions outside the behavior support. This approach mitigates the overestimation problem by using distance-aware pseudo-labels that decay with the distance from the behavior support. The method is shown to outperform existing offline RL methods with minimal tuning on various benchmarks, including D4RL Gym–MuJoCo tasks.

The paper proposes a novel and practical solution to the overestimation problem in offline RL through ASPL, an effective method that augments training with pseudo-labels for OOD actions. The experimental results on D4RL benchmarks demonstrate that ASPL performs
competitively with minimal hyperparameter tuning. However, while the method is shown to work well empirically, the theoretical foundations could benefit from further elaboration, particularly regarding the behavior-aware weighting mechanism. Additionally, a more thorough analysis of the method's generalization across diverse RL tasks would strengthen the claims. The paper provides solid empirical evidence but lacks sufficient theoretical grounding in some areas.

**Strengths:**

Simple and effective solution to overestimation in offline RL:
1. The ASPL method tackles the challenge of sparse coverage in the action space, effectively providing training signals for OOD actions (Sec. 4.1; Eq. 6). This addresses a critical issue in offline RL and improves performance without adding complexity to existing actor–critic pipelines.
2. The integration of pseudo-labeling into the critic update is minimalistic yet effective, requiring no additional actor-side constraints or auxiliary networks (Sec. 4.3). This simplicity enhances the method's appeal for real-world applications where computational efficiency is important.

Empirical validation on D4RL benchmarks:
1. ASPL consistently outperforms several strong offline RL baselines, including TD3+BC, CQL, and IQL, across various D4RL Gym–MuJoCo tasks (Sec. 5.1; Table 1). These results demonstrate the method's robustness and its ability to achieve stable training and competitive performance with minimal hyperparameter tuning.
2. The paper includes detailed experiments showing the sensitivity of ASPL to key hyperparameters like the number of random actions sampled (N) and the pseudolabeling coefficient (α), highlighting its resilience to hyperparameter changes (Sec.5.2; Fig. 3, Fig. 4).

Behavior-aware weighting mechanism:
1. The dynamic adjustment of pseudo-label weight based on behavior coverage (Eq. 9) is a unique and promising feature. This ensures that updates are Bellman-dominated in well-supported regions and pseudo-label-dominated in unsupported regions, improving stability and mitigating extrapolation errors (Sec. 4.2).
2. The simplicity of the behavior-aware weight mechanism—decreasing the weight with increasing behavior coverage—makes the method both effective and easy to integrate into existing RL pipelines (Sec. 4.2).

Minimal tuning burden:
1. ASPL reduces the sensitivity to hyperparameters, particularly the pseudo-label weight α, which simplifies the tuning process compared to other methods (Sec. 5.2). This is a significant advantage for practitioners who need reliable methods with minimal configuration effort.

**Weaknesses:**

Limited theoretical analysis of the behavior-aware weight mechanism:
1. While the paper introduces a behavior-aware weight (Eq. 9), the theoretical explanation and formal analysis of this weight mechanism are not fully developed.
2. The implications of this dynamic weighting, especially in complex environments, could be better justified with more rigorous theoretical analysis (Sec. 4.2). No direct evidence found in the manuscript.
The paper could benefit from a more detailed explanation of why the specific form of distance-aware pseudo-labeling works well across various tasks and datasets. Whilethe empirical results are strong, further formalism in the explanation would add to the method's credibility (Sec. 4.2).

Comparative analysis with more baselines:
1. Although the paper compares ASPL against several state-of-the-art offline RL algorithms, the set of baselines could be expanded to include more diverse approaches, especially methods that use ensemble critics or model-based planning (Sec. 5.1). This would provide a more comprehensive comparison and further demonstrate ASPL's advantages.
2. The paper mainly focuses on tasks in the Gym–MuJoCo environment. Evaluating ASPL on a broader range of tasks, including those with more complex action spaces or other domains like NLP or robotics, would offer insights into the generalizability of
the method (Sec. 5.1). No direct evidence found in the manuscript.

Insufficient discussion of generalization to other action spaces:
1. While the method performs well in continuous action spaces, its applicability to discrete or mixed action spaces is not fully addressed (Sec. 6). A more detailed discussion on how ASPL could be adapted to different action spaces or other types of
decision-making environments would be valuable, particularly for applications beyond Gym–MuJoCo tasks.

Clarifications needed for pseudo-labeling process:
1. The pseudo-labeling process involves sampling random actions from the action space and using distance-based Q-values as pseudo-targets. However, the handling of these pseudo-labels could be explained more clearly, particularly regarding the distance metric and its effects on training dynamics (Eq. 7). Further clarification of how these pseudo-labels interact with the Bellman backup process would improve understanding and reproducibility (Sec. 4.1).

**Questions:**

Provide deeper theoretical analysis of the behavior-aware weight:
1. Elaborate on the formal properties of the behavior-aware weight mechanism and its effects on training stability and performance. A more thorough mathematical analysis could clarify the behavior-aware weight's role and its relationship with other regularization techniques (Sec. 4.2).
2. Consider formalizing the convergence properties of ASPL to help justify its effectiveness and robustness.

Expand baseline comparisons:
1. Include additional baselines that use ensemble critics, model-based approaches, or other advanced techniques in offline RL. This will provide a clearer understanding of how ASPL compares to a broader range of methods (Sec. 5.1).
2. Also, consider evaluating ASPL on tasks with more complex action spaces, such as discrete action tasks or multi-agent settings, to explore its generalization beyond continuous action spaces (Sec. 5.1).

Address generalization to other action spaces:
1. Discuss and provide potential extensions of ASPL for discrete or mixed action spaces. A clear discussion on how the method can be adapted for other action representations, such as categorical or structured actions, would help broaden its applicability (Sec. 6).

Clarify the pseudo-labeling process:
1. Provide a more detailed explanation of the pseudo-labeling process, particularly the distance metric used to calculate the pseudo-Q targets. Clarifying how the pseudolabeling interacts with the Bellman backup would help in understanding its impact on training dynamics (Sec. 4.1).

---

### Meta-Review · Area_Chair_sySx · 2026-01-06

**Summary:**

This paper proposes Action-Space Pseudo-Labeling (ASPL) for offline RL, adding pseudo Q-target supervision for randomly sampled actions that decays with distance to the behavior support to mitigate OOD overestimation. Reviewers generally agree the method is simple and practically motivated, but the decision is driven by concerns about (i) limited novelty/unclear differentiation from conservative value regularization and prior pseudo-label/OOD-value methods (e.g., CQL-like, MCQ/SVR/OAC-BVR style ideas), (ii) insufficient and potentially unreliable empirical evidence (Gym–MuJoCo only, missing harder benchmarks like AntMaze/Kitchen/Adroit, missing std across seeds, baseline inconsistencies), and (iii) lack of clarity/analysis for key design choices (behavior-support distance estimation/scaling, adaptive vs fixed weighting ablations, possible bias propagation through Bellman backups, and at least one reported equation needing clarification).

**Reviewer Concerns:**

There is no author rebuttal provided in the material above, so none of the major concerns can be considered resolved: missing broader benchmarks, missing direct comparisons to closest prior work (MCQ/SVR/OAC-BVR), weak/contested novelty vs CQL-family regularization, unclear distance metric and scaling sensitivity, missing variance reporting and evaluation protocol details, and serious reproducibility doubts due to baseline discrepancies (e.g., SCAL halfcheetah-medium-expert) and nonstandard batch size without justification/ablation.

**Reviewer Scores:**

Given no rebuttal/updates shown, I would not expect meaningful score changes: iaNB likely stays 4→4, BypT stays 4→4, tqHU stays 2→2, gcNK stays 2→2. Overall, the discussion would remain negative-to-borderline with multiple reject scores anchored by empirical and novelty concerns.

---

### Decision · Program_Chairs · 2026-01-26

Reject